# Thrombosis and thrombocytopenia after vaccination against and infection with SARS-CoV-2 in the United Kingdom

Population-based studies can provide important evidence on the safety of COVID-19 vaccines. Using data from the United Kingdom, here we compare observed rates of thrombosis and thrombocytopenia following vaccination against SARS-CoV-2 and infection with SARS-CoV-2 with background (expected) rates in the general population. First and second dose cohorts for ChAdOx1 or BNT162b2 between 8 December 2020 and 2 May 2021 in the United Kingdom were identified. A further cohort consisted of people with no prior COVID-19 vaccination who were infected with SARS-Cov-2 identified by a first positive PCR test between 1 September 2020 and 2 May 2021. The fourth general population cohort for background rates included those people in the database as of 1 January 2017. In total, we included 3,768,517 ChAdOx1 and 1,832,841 BNT162b2 vaccinees, 401,691 people infected with SARS-CoV-2, and 9,414,403 people from the general population. An increased risk of venous thromboembolism was seen after first dose of ChAdOx1 (standardized incidence ratio: 1.12 [95% CI: 1.05 to 1.20]), BNT162b2 (1.12 [1.03 to 1.21]), and positive PCR test (7.27 [6.86 to 7.72]). Rates of cerebral venous sinus thrombosis were higher than otherwise expected after first dose of ChAdOx1 (4.14 [2.54 to 6.76]) and a SARS-CoV-2 PCR positive test (3.74 [1.56 to 8.98]). Rates of arterial thromboembolism after vaccination were no higher than expected but were increased after a SARS-CoV-2 PCR positive test (1.39 [1.21 to 1.61]). Rates of venous thromboembolism with thrombocytopenia were higher than expected after a SARS-CoV-2 PCR positive test (5.76 [3.19 to 10.40]).

Vaccines against SARS-CoV-2 have been developed rapidly using a number of platforms. The ChAdOx1 nCoV-19 (Oxford–AstraZeneca; ChAdOx1) and BNT162b2 mRNA (Pfizer–BioNTech; BNT162b2) vaccines received approval for use in the United Kingdom on 8 and 31 December 2020, respectively. Evidence from clinical trials and real-world data has shown these vaccines to be highly effective in preventing symptomatic COVID-19, severe disease, and hospitalization[1–5].

As COVID-19 vaccines have been approved under emergency authorization, they must continue to be monitored to assess their safety. Instances of rare adverse events have been identified alongside the ongoing nationwide immunization programs[6–8]. A particular concern has arisen regarding thrombotic events, with concurrent thrombocytopenia reported among individuals vaccinated with adenovirus-based vaccines against SARS-CoV-2. As of 26 May 2021, 348 spontaneous reports of major thromboembolic events with thrombocytopenia had been documented following 24 million first doses and 13 million second doses of the ChAdOx1 vaccine in the UK[9]. Although fewer concerns have been raised about safety signals for BNT162b2, instances of immune thrombocytopenia have also been observed among recipients of this vaccine[10].

✉ e-mail: daniel.prietoalhambra@ndorms.ox.ac.uk

In this study, we estimated the incidence of thrombosis, thrombocytopenia, and thrombosis with thrombocytopenia over the 28 days following a first dose of the ChAdOx1 and BNT162b2 vaccines and compared these rates with historical, pre-pandemic rates in the general population. To provide additional context, we also studied the rates of these events after a positive polymerase chain reaction (PCR) test for SARS-CoV-2.

## Results

### Study participants

We included 3,768,517 people vaccinated with ChAdOx1, 1,832,841 people vaccinated with BNT162b2, 401,691 people infected with SARS-CoV-2, and 9,414,403 people from the general population. The vaccinated populations were older, more often female, and had a higher prevalence of all studied comorbidities than the general population. Those infected with SARS-CoV-2 were younger than the general population. Detailed characteristics for all four cohorts are given in Table 1, and stratified by age in the Supplementary Table 1. BNT162b2 vaccination started earlier than ChAdOx1 vaccination. The numbers vaccinated with the two vaccines became similar in late January 2021, and ChAdOx1 vaccines predominated from March to May 2021 (Fig. 1).

### Main results

An increased risk of venous thromboembolism (VTE) was seen after first dose of ChAdOx1 and BNT162b2. For ChAdOx1 first dose, 771 events were expected based on background rates but 866 were observed. For BNT162b2 first dose, 533 events were expected but 595 were observed. These equated to standardized incidence ratios (SIRs) of 1.12 (95% CI: 1.05 to 1.20) and 1.12 (1.03 to 1.21), respectively. Rates were not higher than expected after second dose of either vaccine, see Table 2 and Fig. 2. Meanwhile for those with SARS-CoV-2 PCR positive test, 1090 occurrences of VTE were observed compared to the 150 that would otherwise be expected, with a SIR of 7.27 (6.86 to 7.72). For all study cohorts, incidence of VTE increased with age, see Fig. 3. Among first dose recipients of ChAdOx1 and BNT162b2, incidence rate ratios (IRRs) were seen to be increased among younger age groups, Fig. 4. For people with a SARS-CoV-2 PCR positive test, IRRs were above one for all age groups.

Higher than expected rates of VTE after first dose of ChAdOx1 and BNT162b2 were primarily driven by pulmonary embolism (PE) events rather than occurrences of deep vein thrombosis (DVT), Table 2 and Fig. 2. Meanwhile higher than expected rates of VTE after a SARS-CoV-2 PCR positive test were due to higher rates of DVT and, to an even greater extent, PE. Indeed, the SIR for PE after a SARS-CoV-2 PCR positive test was 12.77 (11.95 to 13.64). The characteristics of study participants with a PE are summarised in the Supplementary Table 5. It can be seen, for example, that the average age of those with such an event after ChAdOx1 and BNT162b2 first dose was 67 and 73, respectively, which compared to an average age of 68 for the general population. Moreover, over 60% of those with a PE after vaccination had at least one related condition or prior medication use.

Rates of cerebral venous sinus thrombosis (CVST) were higher than otherwise expected after first dose of ChAdOx1 and a SARS-CoV-2 PCR positive test. For ChAdOx1 first dose, 16 events observed which compared to 4 expected (SIR: 4.14 [2.54 to 6.76]). For SARS-CoV-2 PCR positive test, 5 events were observed which compared to 1 expected (3.74 [1.56 to 8.98]). Splanchnic vein thrombosis (SVT) for SARS-CoV-2 PCR positive test, with 8 observed compared to the 3 expected (2.81 [1.40 to 5.61]). The average age of those with such an event after ChAdOx1 first dose was 47, which compared to an average age of 48 for the general population (see Supplementary Information).

Rates of arterial thromboembolism (ATE) after vaccination were no higher than expected but were increased after a SARS-CoV-2 PCR positive test, Fig. 2. While 134 ATE would otherwise be expected for the SARS-CoV-2 PCR positive test cohort, 186 events were observed (1.39 [1.21 to 1.61]). This increased risk was particularly pronounced among those aged between 50 and 79 and was primarily driven by an increased risk of myocardial infarction, see Fig. 4.

Thrombocytopenia was more common than expected for all study cohorts, see Table 2 and Fig. 2. SIRs ranged from 1.27 (1.21 to 1.34) for BNT162b2 first dose to 1.47 (1.38 to 1.57) for ChAdOx1 second dose. For all cohorts, rates of thrombocytopenia increased with age, Fig. 3. For first dose ChAdOx1 and BNT162b2 and for second dose ChAdOx1, IRRs were highest among younger age groups, Fig. 4. The average age of those with thrombocytopenia was 66 after ChAdOx1 first dose and 77 after BNT162b2 first dose, which compared to 65 in the general population (see Supplementary Information). As with PE, more than 60% of those with thrombocytopenia after a first dose of ChAdOx1 or BNT162b2 had at least one related condition or prior medication use. An increased risk was also seen for the more specific definition of immune thrombocytopenia ChAdOx1 with a SIR of 1.79 (1.33 to 2.39), while for first dose the SIR was 1.28 (0.83 to 1.96).

More occurrences of VTE with concurrent thrombocytopenia were observed than expected for ChAdOx1 first dose, 16 compared to 12, although the SIR confidence interval crossed one (1.38 [0.85 to 2.26]). Meanwhile, for the SARS-CoV-2 PCR positive test cohort, 11 events were observed which compared to the two expected (5.76 [3.19 to 10.40]). No more events than expected were seen for ATE with thrombocytopenia.

### Sensitivity analyses

Results from sensitivity analyses were generally consistent with the results from the primary analyses (see Supplementary Information). All primary and sensitivity analysis results are available in an interactive web application: https://livedataoxford.shinyapps.io/CovidVaccinationSafetyStudy/.

## Discussion

In a cohort of 5.6 million people vaccinated against SARS-CoV-2, 157 (0.003%) more than expected had a VTE in the 28 days following their first dose. Thrombocytopenia was also more common after vaccination, with 1145 (0.021%) more events seen after a first dose of ChAdOx1 or BNT162b2 than would be expected. Rates of CVST were also higher than expected after a fist dose of ChAdOx1 with 12 more events than the 4 that would typically be expected, which equated to a standardized event difference proportion of 0.0003%. Meanwhile, among close to 400,000 people who were not yet vaccinated and who had a positive SARS-CoV-2 PCR positive test, 940 (0.23%) more people had a VTE in the 90 days after their positive test, 151 (0.0027%) had thrombocytopenia, and 4 (0.001%) had CVST. Moreover, this SARS-CoV-2 PCR cohort also had an increased risk of arterial thromboembolism with 53 (0.01%) more cases observed than would normally be expected. Study participants with a positive SARS-CoV-2 PCR positive test were also at increased risk of a VTE with thrombocytopenia with 9 (0.002%) more events observed than would be otherwise expected.

Concerns over thrombosis—alone and with thrombocytopenia—have been raised from spontaneous reports data since March 2021[6,7,11]. Case series have been published, suggesting a new clinical entity known as vaccine-induced immune thrombotic thrombocytopenia (VITT), presenting as unusual thrombosis with raised antibodies against platelet factor 4. To date, thrombosis and thrombocytopenia has primarily been a concern for adenoviral-based vectors[12]. However, in our study we have seen comparable safety signals for PE and thrombocytopenia for both ChAdOx1 and BNT162b. Thrombocytopenia post vaccination has previously been reported after receipt of

**Table 1 | Characteristics of study participants**

| | Vaccinated with ChadOx1 first dose | Vaccinated with ChadOx1 second dose | Vaccinated with BNT162b2 first dose | Vaccinated with BNT162b2 second dose | Infected with SARS-CoV-2 | General population |
|---|---|---|---|---|---|---|
| *N* | 3,768,517 | 1,091,660 | 1,832,841 | 1,301,994 | 401,691 | 9,414,403 |
| Age | 56 [47 to 67] | 71 [60 to 76] | 65 [50 to 77] | 71 [53 to 80] | 42 [31 to 54] | 48 [34 to 63] |
| Age: 20 to 29 | 172,501 (4.6%) | 31,844 (2.9%) | 100,417 (5.5%) | 64,580 (5.0%) | 89,849 (22.4%) | 1,529,282 (16.2%) |
| Age: 30 to 39 | 289,137 (7.7%) | 48,581 (4.5%) | 146,784 (8.0%) | 96,100 (7.4%) | 89,156 (22.2%) | 1,699,641 (18.1%) |
| Age: 40 to 49 | 652,820 (17.3%) | 68,858 (6.3%) | 199,485 (10.9%) | 118,705 (9.1%) | 81,969 (20.4%) | 1,675,732 (17.8%) |
| Age: 50 to 59 | 1,102,767 (29.3%) | 122,937 (11.3%) | 295,264 (16.1%) | 162,867 (12.5%) | 78,377 (19.5%) | 1,671,410 (17.8%) |
| Age: 60 to 69 | 813,677 (21.6%) | 225,941 (20.7%) | 328,039 (17.9%) | 181,687 (14.0%) | 39,543 (9.8%) | 1,255,103 (13.3%) |
| Age: 70 to 79 | 558,629 (14.8%) | 445,900 (40.8%) | 375,954 (20.5%) | 333,504 (25.6%) | 14,312 (3.6%) | 960,021 (10.2%) |
| Age: 80 or older | 178,986 (4.7%) | 147,599 (13.5%) | 386,898 (21.1%) | 344,551 (26.5%) | 8,485 (2.1%) | 623,214 (6.6%) |
| Sex: Male | 1,823,627 (48.4%) | 481,252 (44.1%) | 765,366 (41.8%) | 515,665 (39.6%) | 184,816 (46.0%) | 4,697,418 (49.9%) |
| Years of prior observation time | 16.1 [6.7 to 27.7] | 20.5 [8.2 to 32.4] | 18.3 [7.1 to 30.5] | 19.4 [7.6 to 31.5] | 11.5 [4.5 to 22.5] | 12.7 [4.9 to 24.0] |
| Comorbidities | | | | | | |
| Autoimmune disease | 104,916 (2.8%) | 47,545 (4.4%) | 68,272 (3.7%) | 50,640 (3.9%) | 7982 (2.0%) | 196,057 (2.1%) |
| Antiphospholipid syndrome | 2419 (0.1%) | 855 (0.1%) | 1456 (0.1%) | 890 (0.1%) | 224 (0.1%) | 4134 (0.0%) |
| Thrombophilia | 6132 (0.2%) | 1909 (0.2%) | 3516 (0.2%) | 2106 (0.2%) | 617 (0.2%) | 10,814 (0.1%) |
| Asthma | 559,364 (14.8%) | 166,446 (15.2%) | 288,162 (15.7%) | 191,066 (14.7%) | 63,452 (15.8%) | 1,258,204 (13.4%) |
| Atrial fibrillation | 110,915 (2.9%) | 76,063 (7.0%) | 121,439 (6.6%) | 99,983 (7.7%) | 5555 (1.4%) | 231,355 (2.5%) |
| Malignant neoplastic disease | 319,455 (8.5%) | 185,659 (17.0%) | 278,320 (15.2%) | 224,378 (17.2%) | 15,571 (3.9%) | 606,023 (6.4%) |
| Diabetes mellitus | 382,759 (10.2%) | 181,179 (16.6%) | 286,916 (15.7%) | 190,137 (14.6%) | 29,718 (7.4%) | 684,076 (7.3%) |
| Obesity | 190,490 (5.1%) | 67,126 (6.1%) | 104,647 (5.7%) | 66,988 (5.1%) | 18,783 (4.7%) | 347,303 (3.7%) |
| Heart disease | 412,183 (10.9%) | 232,123 (21.3%) | 364,325 (19.9%) | 279,390 (21.5%) | 25,270 (6.3%) | 824,090 (8.8%) |
| Hypertensive disorder | 914,963 (24.3%) | 449,674 (41.2%) | 665,230 (36.3%) | 513,359 (39.4%) | 54,133 (13.5%) | 1,761,190 (18.7%) |
| Renal impairment | 242,054 (6.4%) | 157,370 (14.4%) | 244,363 (13.3%) | 202,362 (15.5%) | 13,896 (3.5%) | 511,751 (5.4%) |
| Chronic Obstructive Pulmonary Disease | 119,282 (3.2%) | 78,053 (7.1%) | 98,131 (5.4%) | 77,273 (5.9%) | 5177 (1.3%) | 233,441 (2.5%) |
| Dementia | 45,477 (1.2%) | 31,306 (2.9%) | 36,820 (2.0%) | 30,967 (2.4%) | 4052 (1.0%) | 95,934 (1.0%) |
| Medication use (183 days prior to four days prior) | | | | | | |
| Non-steroidal anti-inflammatory drugs | 413,568 (11.0%) | 167,098 (15.3%) | 257,194 (14.0%) | 188,258 (14.5%) | 37,849 (9.4%) | 1,212,845 (12.9%) |
| Cox2 inhibitors | 3138 (0.1%) | 1144 (0.1%) | 1657 (0.1%) | 1186 (0.1%) | 273 (0.1%) | 5902 (0.1%) |
| Systemic corticosteroids | 186,909 (5.0%) | 77,560 (7.1%) | 113,752 (6.2%) | 83,783 (6.4%) | 16,920 (4.2%) | 537,367 (5.7%) |
| Antithrombotic and anticoagulant therapies | 70,586 (1.9%) | 43,189 (4.0%) | 66,851 (3.6%) | 52,539 (4.0%) | 3827 (1.0%) | 190,056 (2.0%) |
| Lipid modifying agents | 137,927 (3.7%) | 71,960 (6.6%) | 105,106 (5.7%) | 78,938 (6.1%) | 8267 (2.1%) | 290,693 (3.1%) |
| Antineoplastic and immunomodulating agents | 39,587 (1.1%) | 14,455 (1.3%) | 27,003 (1.5%) | 20,039 (1.5%) | 7697 (1.9%) | 161,422 (1.7%) |
| Hormonal contraceptives for systemic use | 61,409 (1.6%) | 10,464 (1.0%) | 30,756 (1.7%) | 20,421 (1.6%) | 13,409 (3.3%) | 240,743 (2.6%) |
| Tamoxifen | 1281 (0.0%) | 446 (0.0%) | 817 (0.0%) | 533 (0.0%) | 85 (0.0%) | 2750 (0.0%) |
| Sex hormones and modulators of the genital system | 109,790 (2.9%) | 25,744 (2.4%) | 55,925 (3.1%) | 39,096 (3.0%) | 16,663 (4.1%) | 304,728 (3.2%) |
| Summary count of conditions and medications of interest | | | | | | |
| One or more condition of interest | 1,029,813 (27.3%) | 503,797 (46.1%) | 784,075 (42.8%) | 574,618 (44.1%) | 72,933 (18.2%) | 1,956,104 (20.8%) |
| One or more medication of interest | 576,743 (15.3%) | 218,008 (20.0%) | 346,629 (18.9%) | 252,789 (19.4%) | 58,600 (14.6%) | 1,650,421 (17.5%) |
| One or more condition/medication of interest | 1,385,947 (36.8%) | 603,396 (55.3%) | 951,649 (51.9%) | 692,584 (53.2%) | 114,788 (28.6%) | 3,062,267 (32.5%) |

Characteristics of the participants in the four study cohorts used for the primary analyses. Participants were aged 20 years or older and had at least one year of prior history before index date in the database. Those in the general population cohort were present in the database as of 1st January 2017. People infected with SARS-CoV-2 had a confirmatory positive RT-PCR test.
*Conditions of interest: autoimmune disease, antiphospholipid syndrome, thrombophilia, asthma, atrial fibrillation, malignant neoplastic disease, diabetes mellitus, obesity, or renal impairment.
†Medications of interest: non-steroidal anti-inflammatory drugs, Cox2 inhibitors, systemic corticosteroids, hormonal contraceptives, tamoxifen, and sex hormones and modulators of the genital system.

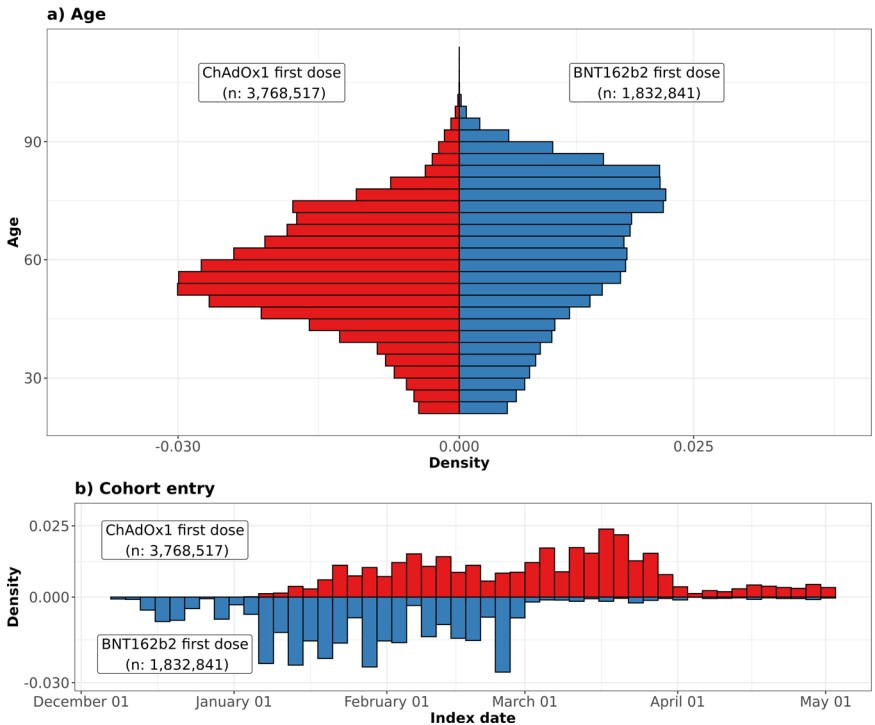

**Fig. 1 | Distribution of age profiles and date of cohort entry among people vaccinated against SARS-CoV-2.** Distribution of ChAdOx1 (red) and BNT162b2 vaccine recipients (blue) by age (**a**) and calendar time (**b**).

other vaccines, such as those against influenza[13], measles, mumps, and rubella[14], and hepatitis B[15]. Consequently, the finding from our study is in line with the existing observation that vaccines, as well as infections, initiate immune-mediated mechanisms that can induce protective immunity but may also lead to an autoimmune response[16]. Indeed, we see in our study a similar increase in risk for thrombocytopenia among COVID-19 cases.

A study of 280,000 vaccinees aged between 18 and 65 in Denmark and Norway assessed the 28-day incidence rates of thromboembolic events and coagulation disorders following ChAdOx1[17]. Similar to our analyses, Pottegård et al applied a historical comparator design with indirect standardization. They found a 2-fold increased rate of VTE, an 80% increase in rates of PE, and a 20-fold increased rate of CVST. The authors also reported a 3-fold higher-than-expected rate of thrombocytopenia. As in our study, they observed similar rates of arterial events among those vaccinated as would be expected given rates in the general population.

Another study using data from the UK, which used a self-controlled case series analysis approach, found ChAdOx1 to be associated with an increased risk of thrombocytopenia, VTE, and CVST, while BNT162b was associated with increased risks of ATE, ischaemic stroke, and CVST[18]. Meanwhile, a nested case-control study from Scotland found no increase in risk of VTE with either vaccine[19]. However, the authors of this latter study also reported potential increased risks of ATE and hemorrhagic events with ChAdOx1, although these were not confirmed in subsequent self-controlled case series analysis.

As well as analysing the rates of adverse events, we have also described the characteristics of those persons affected. Study participants with pulmonary embolism and those with thrombocytopenia after vaccination were generally older and more often had a prior history of related conditions or medications, which is similar to the profiles of people with these events in previous years. This is though in contrast to the early case series describing the profiles of patients with vaccine-induced thrombosis with thrombocytopenia, which have

often identified patients aged under 60, more often female, and with relatively few comorbidities described. This may in part be explained by selection biases affecting case series, but may also reflect the broader definitions used in this study. For example, we do not have measurements of anti-PF4 antibodies and so could not use this for defining study outcomes.

Our study has limitations. The time period studied covered the initial phases of vaccination in the UK, when vaccines were prioritized for older, more vulnerable populations and healthcare staff. We therefore saw a higher prevalence of conditions such as asthma and diabetes in those vaccinated than in the general population. Although we used indirect standardization to account for differences in age distributions, remaining residual confounding could explain some of our findings. Such bias could result in overestimated safety signals due to remaining imbalances in the baseline outcome risk when comparing vaccinated and background populations.

Measurement error is unavoidable in studies such as ours based on routinely-collected health care data. However, any errors are likely to have been non-differential across our vaccinated and unvaccinated cohorts and should therefore not have affected our relative rate estimates. As we only used primary care data, we may have underestimated absolute risks due to a lack of hospital linkage. However, previous studies have shown that CPRD captures rare events well, even without linkage to Hospital Episode Statistics[20].

Our study also has strengths. The large sample of 5.6 million vaccinees allowed us to assess very rare events that are generally not observed in clinical trials. While spontaneous reports provide a valuable resource for identifying potential safety signals, population-based studies such as ours allow for further consideration of whether the rates being observed after vaccinations exceed those that would be expected to occur in the absence of any vaccination. We used a well-established source of routinely collected health data previously used for vaccine safety studies[21,22]. Moreover, including cohorts of people infected with SARS-CoV-2 provided much needed context for interpreting our findings.

**Table 2 | Observed versus expected events among people vaccinated against SARS-CoV-2 or with a positive PCR test for SARS-CoV-2**

| | N | Person-years | Observed events | Expected events | Standardized event difference proportion % | SIR (95% CI) |
|---|---|---|---|---|---|---|
| **Thrombosis** | | | | | | |
| **Cerebral venous sinus thrombosis** | | | | | | |
| Vaccinated with ChAdOx1 first dose | 3,764,507 | 277,866 | 16 | 3.9 | 0.0003% | 4.14 (2.54 to 6.76) |
| SARS-CoV-2 PCR positive test | 401,568 | 95,737 | 5 | 1.3 | 0.0009% | 3.74 (1.56 to 8.98) |
| **Deep vein thrombosis** | | | | | | |
| Vaccinated with ChAdOx1 first dose | 3,757,806 | 277,341 | 456 | 448.5 | 0.0002% | 1.02 (0.93 to 1.11) |
| Vaccinated with ChAdOx1 second dose | 1,087,890 | 57,802 | 121 | 141.1 | −0.0018% | 0.86 (0.72 to 1.02) |
| Vaccinated with BNT162b2 first dose | 1,827,104 | 139,108 | 303 | 303.9 | 0.0000% | 1.00 (0.89 to 1.12) |
| Vaccinated with BNT162b2 second dose | 1,298,221 | 86,134 | 182 | 213.6 | −0.0024% | 0.85 (0.74 to 0.99) |
| SARS-CoV-2 PCR positive test | 401,111 | 95,592 | 265 | 91.2 | 0.0433% | 2.91 (2.58 to 3.28) |
| **Pulmonary embolism** | | | | | | |
| Vaccinated with ChAdOx1 first dose | 3,757,618 | 277,328 | 466 | 370.2 | 0.0025% | 1.26 (1.15 to 1.38) |
| Vaccinated with ChAdOx1 second dose | 1,087,821 | 57,800 | 99 | 124.8 | −0.0024% | 0.79 (0.65 to 0.97) |
| Vaccinated with BNT162b2 first dose | 1,826,976 | 139,097 | 324 | 258.3 | 0.0036% | 1.25 (1.12 to 1.40) |
| Vaccinated with BNT162b2 second dose | 1,298,128 | 86,131 | 153 | 182.7 | −0.0023% | 0.84 (0.71 to 0.98) |
| SARS-CoV-2 PCR positive test | 401,143 | 95,487 | 876 | 68.6 | 0.2013% | 12.77 (11.95 to 13.64) |
| **Splanchnic Vein Thrombosis** | | | | | | |
| Vaccinated with ChAdOx1 first dose | 3,764,449 | 277,862 | 17 | 13.9 | 0.0001% | 1.22 (0.76 to 1.97) |
| Vaccinated with ChAdOx1 second dose | 1,090,856 | 57,967 | 5 | 3.7 | 0.0001% | 1.34 (0.56 to 3.21) |
| SARS-CoV-2 PCR positive test | 401,564 | 95,736 | 8 | 2.8 | 0.0013% | 2.81 (1.40 to 5.61) |
| **Venous thromboembolism** | | | | | | |
| Vaccinated with ChAdOx1 first dose | 3,751,401 | 276,841 | 866 | 770.9 | 0.0025% | 1.12 (1.05 to 1.20) |
| Vaccinated with ChAdOx1 second dose | 1,085,037 | 57,645 | 211 | 252 | −0.0038% | 0.84 (0.73 to 0.96) |
| Vaccinated with BNT162b2 first dose | 1,822,927 | 138,779 | 595 | 533.2 | 0.0034% | 1.12 (1.03 to 1.21) |
| Vaccinated with BNT162b2 second dose | 1,295,309 | 85,938 | 324 | 376.9 | −0.0041% | 0.86 (0.77 to 0.96) |
| SARS-CoV-2 PCR positive test | 400,723 | 95,357 | 1,090 | 149.8 | 0.2346% | 7.27 (6.86 to 7.72) |
| **Myocardial infarction** | | | | | | |
| Vaccinated with ChAdOx1 first dose | 3,755,717 | 277,176 | 606 | 708.6 | −0.0027% | 0.86 (0.79 to 0.93) |
| Vaccinated with ChAdOx1 second dose | 1,086,394 | 57,724 | 166 | 238.1 | −0.0066% | 0.70 (0.60 to 0.81) |
| Vaccinated with BNT162b2 first dose | 1,824,272 | 138,887 | 442 | 500.6 | −0.0032% | 0.88 (0.80 to 0.97) |
| Vaccinated with BNT162b2 second dose | 1,296,561 | 86,025 | 283 | 355.8 | −0.0056% | 0.80 (0.71 to 0.89) |
| SARS-CoV-2 PCR positive test | 400,986 | 95,580 | 167 | 114.4 | 0.0131% | 1.46 (1.25 to 1.70) |
| **Ischemic stroke** | | | | | | |
| Vaccinated with ChAdOx1 first dose | 3,762,624 | 277,719 | 128 | 155.8 | −0.0007% | 0.82 (0.69 to 0.98) |
| Vaccinated with ChAdOx1 second dose | 1,089,880 | 57,912 | 47 | 63.3 | −0.0015% | 0.74 (0.56 to 0.99) |
| Vaccinated with BNT162b2 first dose | 1,830,001 | 139,335 | 146 | 132.9 | 0.0007% | 1.10 (0.93 to 1.29) |
| Vaccinated with BNT162b2 second dose | 1,300,162 | 86,264 | 68 | 99.9 | −0.0025% | 0.68 (0.54 to 0.86) |
| SARS-CoV-2 PCR positive test | 401,463 | 95,710 | 28 | 23.2 | 0.0012% | 1.21 (0.83 to 1.75) |
| **Arterial thromboembolism** | | | | | | |
| Vaccinated with ChAdOx1 first dose | 3,752,849 | 276,954 | 712 | 839.2 | −0.0034% | 0.85 (0.79 to 0.91) |
| Vaccinated with ChAdOx1 second dose | 1,084,976 | 57,646 | 209 | 292.9 | −0.0077% | 0.71 (0.62 to 0.82) |
| Vaccinated with BNT162b2 first dose | 1,822,051 | 138,713 | 568 | 616 | −0.0026% | 0.92 (0.85 to 1.00) |
| Vaccinated with BNT162b2 second dose | 1,295,041 | 85,920 | 344 | 443.6 | −0.0077% | 0.78 (0.70 to 0.86) |
| SARS-CoV-2 PCR positive test | 400,818 | 95,538 | 186 | 133.5 | 0.0131% | 1.39 (1.21 to 1.61) |
| **Stroke** | | | | | | |
| Vaccinated with ChAdOx1 first dose | 3,755,737 | 277,180 | 635 | 723.8 | −0.0024% | 0.88 (0.81 to 0.95) |
| Vaccinated with ChAdOx1 second dose | 1,086,113 | 57,695 | 216 | 287.1 | −0.0065% | 0.75 (0.66 to 0.86) |
| Vaccinated with BNT162b2 first dose | 1,824,739 | 138,920 | 537 | 613.3 | −0.0042% | 0.88 (0.80 to 0.95) |
| Vaccinated with BNT162b2 second dose | 1,296,430 | 86,003 | 312 | 460.4 | −0.0114% | 0.68 (0.61 to 0.76) |
| SARS-CoV-2 PCR positive test | 401,062 | 95,608 | 121 | 111.3 | 0.0024% | 1.09 (0.91 to 1.30) |
| **Thrombocytopenia** | | | | | | |
| **Immune thrombocytopenia** | | | | | | |
| Vaccinated with ChAdOx1 first dose | 3,764,198 | 277,841 | 45 | 25.2 | 0.0005% | 1.79 (1.33 to 2.39) |

**Table 2 (continued) | Observed versus expected events among people vaccinated against SARS-CoV-2 or with a positive PCR test for SARS-CoV-2**

| | N | Person-years | Observed events | Expected events | Standardized event difference proportion % | SIR (95% CI) |
|---|---|---|---|---|---|---|
| Vaccinated with ChAdOx1 second dose | 1,090,713 | 57,959 | 8 | 7.7 | 0.0000% | 1.04 (0.52 to 2.07) |
| Vaccinated with BNT162b2 first dose | 1,831,194 | 139,430 | 21 | 16.4 | 0.0003% | 1.28 (0.83 to 1.96) |
| Vaccinated with BNT162b2 second dose | 1,301,033 | 86,326 | 9 | 11.4 | −0.0002% | 0.79 (0.41 to 1.52) |
| SARS-CoV-2 PCR positive test | 401,539 | 95,729 | 10 | 5.9 | 0.0010% | 1.70 (0.92 to 3.17) |
| **Thrombocytopenia** | | | | | | |
| Vaccinated with ChAdOx1 first dose | 3,728,941 | 275,077 | 2615 | 1824.10 | 0.0212% | 1.43 (1.38 to 1.49) |
| Vaccinated with ChAdOx1 second dose | 1,070,404 | 56,790 | 886 | 603 | 0.0264% | 1.47 (1.38 to 1.57) |
| Vaccinated with BNT162b2 first dose | 1,801,502 | 137,106 | 1,653 | 1298.80 | 0.0197% | 1.27 (1.21 to 1.34) |
| Vaccinated with BNT162b2 second dose | 1,277,216 | 84,660 | 1,229 | 926.8 | 0.0237% | 1.33 (1.25 to 1.40) |
| SARS-CoV-2 PCR positive test | 399,239 | 95,143 | 536 | 384.8 | 0.0379% | 1.39 (1.28 to 1.52) |
| **Thrombosis with thrombocytopenia** | | | | | | |
| **Deep vein thrombosis with thrombocytopenia** | | | | | | |
| Vaccinated with ChAdOx1 first dose | 3,764,546 | 277,869 | 11 | 7.4 | 0.0001% | 1.49 (0.83 to 2.69) |
| Vaccinated with BNT162b2 second dose | 1,301,218 | 86,339 | 8 | 3.6 | 0.0003% | 2.24 (1.12 to 4.47) |
| **Pulmonary embolism with thrombocytopenia** | | | | | | |
| Vaccinated with ChAdOx1 first dose | 3,764,563 | 277,870 | 8 | 4.6 | 0.0001% | 1.72 (0.86 to 3.44) |
| SARS-CoV-2 PCR positive test | 401,571 | 95,737 | 9 | 0.8 | 0.0020% | 11.95 (6.22 to 22.97) |
| **Venous thromboembolism with thrombocytopenia** | | | | | | |
| Vaccinated with ChAdOx1 first dose | 3,764,482 | 277,864 | 16 | 11.6 | 0.0001% | 1.38 (0.85 to 2.26) |
| Vaccinated with BNT162b2 first dose | 1,831,401 | 139,447 | 6 | 8.1 | −0.0001% | 0.74 (0.33 to 1.64) |
| Vaccinated with BNT162b2 second dose | 1,301,179 | 86,336 | 10 | 5.8 | 0.0003% | 1.73 (0.93 to 3.22) |
| SARS-CoV-2 PCR positive test | 401,566 | 95,736 | 11 | 1.9 | 0.0023% | 5.76 (3.19 to 10.40) |
| **Arterial thromboembolism with thrombocytopenia** | | | | | | |
| Vaccinated with ChAdOx1 first dose | 3,764,535 | 277,868 | 7 | 11.5 | −0.0001% | 0.61 (0.29 to 1.27) |
| Vaccinated with BNT162b2 first dose | 1,831,410 | 139,447 | 6 | 9.8 | −0.0002% | 0.61 (0.27 to 1.36) |
| **Stroke with thrombocytopenia** | | | | | | |
| Vaccinated with ChAdOx1 first dose | 3,764,565 | 277,870 | 7 | 6.7 | 0.0000% | 1.05 (0.50 to 2.21) |
| Vaccinated with BNT162b2 first dose | 1,831,463 | 139,451 | 5 | 6.2 | −0.0001% | 0.80 (0.33 to 1.93) |

For each event of interest, the number of people contributing to the analysis from the target population, their person-years contributed, and the number of observed events are given. Expected events are estimated using indirect standardization to the general population. Standardized incidence ratios (SIRs) with 95% confidence intervals (CIs) were estimated. Events with fewer than 5 occurrences were omitted for privacy reasons.

In conclusion, in a cohort of 5.6 million people vaccinated against SARS-Cov-2, thrombosis, thrombocytopenia, and thrombosis with thrombocytopenia were very rare events. A similar safety signal was seen for VTE and thrombocytopenia (overall and specifically immune-related) was seen after first dose of both ChAdOx1 and BNT162b2, and of CVST after a first dose of ChAdOx1. Although the occurrence of VTE after vaccination was 1.1-fold above that expected in the general population, among those infected with SARS-CoV-2 it was more than 7-fold the background (expected) rate. Infection with SARS-CoV-2 prior to any vaccination against COVID-19 was also associated with increased risks of thrombocytopenia, arterial thromboembolism, and VTE with thrombocytopenia. These findings underline the relative safety of vaccines compared to the numerous ill-effects of being infected by SARS-CoV-2 for those people that remain unvaccinated.

## Methods
### Study design, setting, and data sources
People vaccinated against SARS-CoV-2, people infected with SARS-CoV-2, and a background cohort to estimate pre-pandemic background rates were identified from Clinical Practice Research Datalink (CPRD) AURUM. The use of CPRD data was approved by the Independent Scientific Advisory Committee (21_000391 and 20_000211). CPRD AURUM is an established primary care databases broadly

representative of the UK population[13,14], and previous research has demonstrated its validity for vaccine safety surveillance[15,16]. The database was mapped to the Observational Medical Outcomes Partnership (OMOP) Common Data Model (CDM)[17].

### Study participants and follow-up
Six cohorts were studied. Four vaccination cohorts were identified which included people vaccinated with either ChAdOx1 or BNT162b2 for either their first or second dose between 8 December 2020 and 2 May 2021. They were followed for up to 28 days from their first vaccination (index date). A third cohort consisted of people newly infected with SARS-Cov-2 identified by a first positive RT-PCR test between 1 September 2020 and 2 May 2021. The test date was used as the index date. They were followed for up to 90 days. The fourth cohort, a general population background cohort, included people in the database as of 1 January 2017. Follow-up for this cohort ran up to 31 December 2019.

All participants were required to be aged 20 years or older and, for the primary analysis, have at least 1 year of prior history available. Participants did not contribute to an analysis if they had the same event recorded in the year before their index date. Time at risk was censored if an individual had the outcome of interest or exited the database before the end of follow-up.

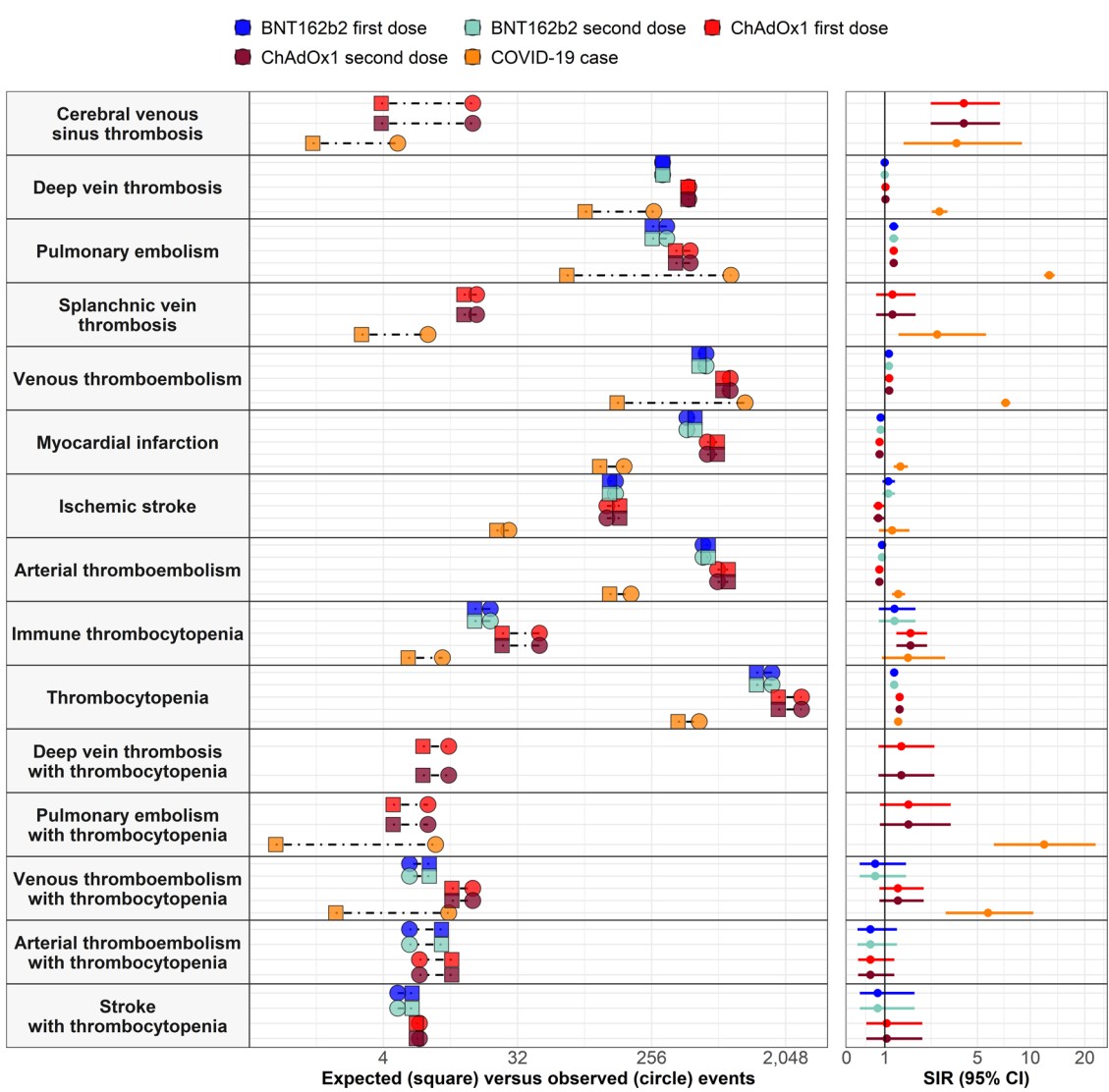

**Fig. 2 | Background and post-vaccine rates of thromboembolic events and thrombocytopenia by age.** Events with less than 5 occurrences have been omitted for privacy reasons. Point estimates with 95% confidence intervals.

One sensitivity analysis was conducted removing the requirements of a year of prior history for all study cohorts. A second sensitivity analysis was conducted where the background population entered the study cohort on the date of their first primary care visit or contact between 2017 and 2019.

### Study outcomes

We used diagnostic codes to identify five venous thromboembolic events: cerebral venous sinus thrombosis (CVST), deep vein thrombosis (DVT), pulmonary embolism (PE), splanchnic vein thrombosis (SVT), and the composite event venous thromboembolism (VTE), which encompassed DVT and PE. We identified two arterial thromboembolic events (ATE): myocardial infarction and ischemic stroke. We also used an overall stroke definition that included non-specific, hemorrhagic, and ischemic stroke codes.

Thrombocytopenia was identified using diagnostic codes and laboratory data showing platelets count between 10,000 and 150,000 platelets per microliter, based on the Brighton collaboration definition[18]. Immune thrombocytopenia was identified using diagnostic codes.

Thrombosis with thrombocytopenia syndromes (TTS) was identified where thrombocytopenia was observed within 10 days before or after thrombosis. This time window was broadened in a sensitivity analysis.

Results for additional related outcomes (intestinal infarction, platelet disorder, portal vein thrombosis, and thrombocytopenic purpura) are reported in the Supplementary Information.

### Statistical methods

For each cohort and outcome of interest, we describe the cohort's age, sex, comorbidities, and medication use within 6 months before and up to 4 days before the index date. We report the number of events observed and crude incidence rates per 100,000 person-years with 95% confidence intervals (CIs). We used indirect standardization with the background cohort as the standard to estimate the number of events expected for the vaccination and SARS-CoV-2 cohorts if their risk was the same as that of the general population[19]. We estimated standardized incidence ratios (SIRs) and 95% confidence intervals comparing observed and expected rates. We stratified all analyses by 10-year age bands and sex and analyses of those vaccinated by calendar month. We calculated the standardized event difference proportion to provide a measure of absolute risk. To avoid re-identification, we do not report any analysis with under 5 cases.

### Reporting summary

Further information on research design is available in the Nature Portfolio Reporting Summary linked to this article.

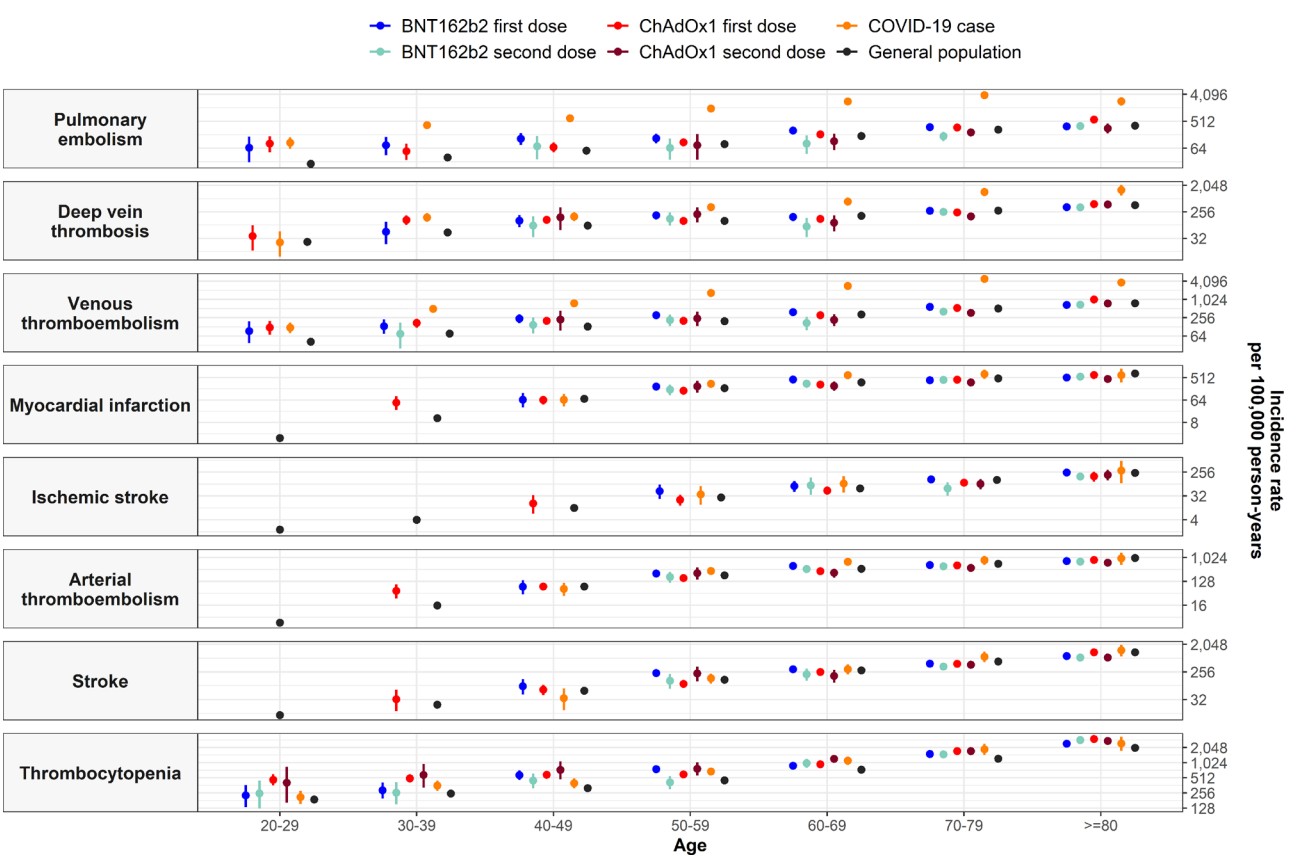

**Fig. 3 | Incidence rate ratios (IRRs) for thromboembolic events and thrombocytopenia by age.** Events with less than 5 occurrences have been omitted for privacy reasons. Point estimates with 95% confidence intervals.

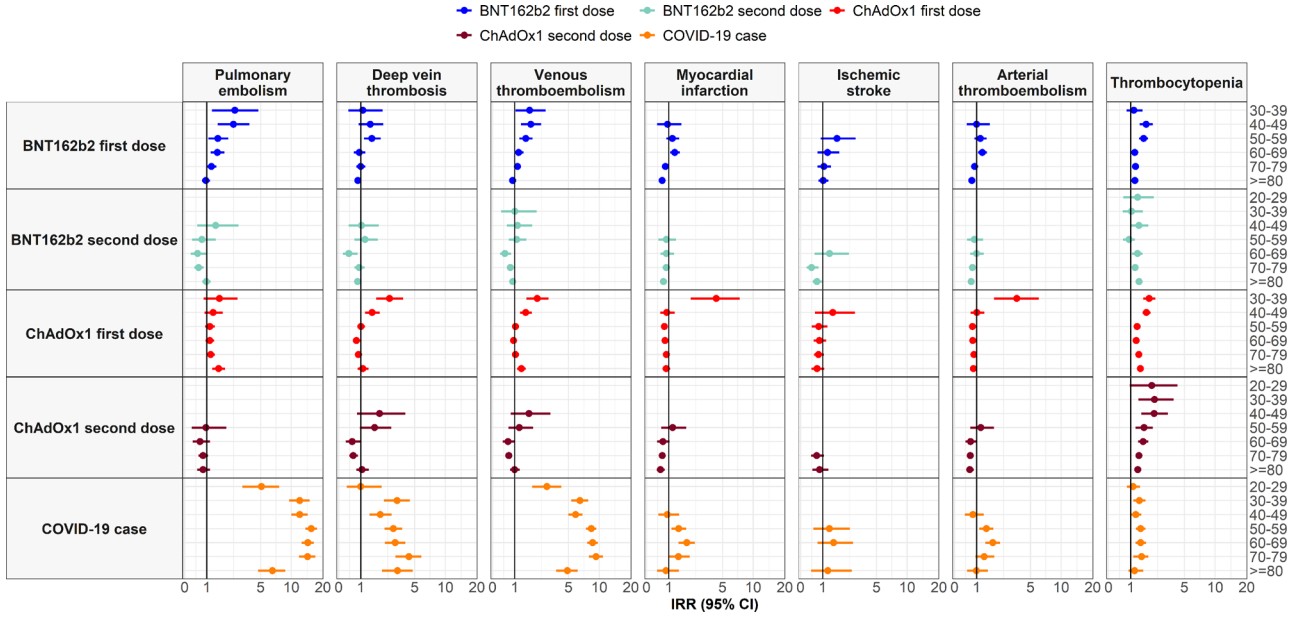

**Fig. 4 | Expected versus observed events among those vaccinated against SARS-CoV-2 and those with a SARS-CoV-2 infection.** Expected events for each of the study cohorts based on indirect standardization using rates from the general population between 2017 and 2019 are compared with the number of observed events seen in each cohort on the panels on the left. Corresponding standardized incidence ratios (SIRs) with 95% confidence intervals (95% CI) are shown in in the panels on the right.

## Code availability

The analytic code to perform the study is available at https://github.com/oxford-pharmacoepi/CovidVaccinationSafetyStudy (https://doi.org/10.5281/zenodo.6584004).

## Data availability

Patient level data used in this study was obtained through an approved application to the Clinical Practice Research Datalink (CPRD) AURUM (application number 21_000391) and is only available following an approval process in order to safeguard the confidentiality of patient data. Details on how to apply for data access can be found at https://cprd.com/data-access. Aggregated data for Figs. 2, 3 and 4 are provided as source data for this paper. Source data are provided with this paper.

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

## Acknowledgements

This study was funded by the European Medicines Agency in the form of a competitive tender (Lot ROC No EMA/2017/09/PE). We acknowledge Prof Johan Van der Lei for the overall management of this research grant. This document expresses the opinion of the authors of the paper, and may not be understood or quoted as being made on behalf of or reflecting the position of the European Medicines Agency or one of its committees or working parties. E.R. was supported by Instituto de Salud Carlos III (grant number CM20/00174). D.P.A. is funded through a National Institute for Health Research (NIHR) Senior Research Fellowship (Grant number SRF-2018-11-ST2-004).

## Author contributions

E.B., X.L., V.Y.S., and D.P.A. led study design. A.D., N.J., and T.D.S. led data collection and processing. C.R., E.M.H., E.M., and K.V. provided clinical input and contributed to the identification of study outcomes. P.R. led the coordination of the project and contracting. E.B. and D.P.A. led the drafting of the manuscript. All authors were involved in the interpretation of the results, and the critical review and approval of the manuscript.

## Competing interests

D.P.A.'s research group has received research grants from the European Medicines Agency, from the Innovative Medicines Initiative, from Amgen, Chiesi, and from UCB Biopharma; and consultancy or speaker fees from Astellas, Amgen and UCB Biopharma. The remaining authors declare no competing interests.

## Additional information

Edward Burn [1,2,6], Xintong Li [2,6], Antonella Delmestri [2], Nathan Jones[2], Talita Duarte-Salles [1], Carlen Reyes[1],
Eugenia Martinez-Hernandez [3], Edelmira Marti [4], Katia M. C. Verhamme[5], Peter R. Rijnbeek [5], Victoria Y. Strauss[2,7] &
Daniel Prieto-Alhambra [2,5,7] ✉

[1]Fundació Institut Universitari per a la recerca a l'Atenció Primària de Salut Jordi Gol i Gurina (IDIAPJGol), Barcelona, Spain. [2]Centre for Statistics in Medicine
(CSM), Nuffield Department of Orthopaedics, Rheumatology and Musculoskeletal Sciences (NDORMS), University of Oxford, Oxford, UK. [3]Department of
Neurology, Hospital Clinic and University of Barcelona, Barcelona, Spain. [4]Hemostasis and Thrombosis Unit, Hematology Department, Hospital Clínico
Universitario de Valencia, Valencia, Spain. [5]Department of Medical Informatics, Erasmus University Medical Center, Rotterdam, The Netherlands. [6]These
authors contributed equally: Edward Burn, Xintong Li. [7]These authors jointly supervised this work: Victoria Y. Strauss, Daniel Prieto-Alhambra.
✉e-mail: daniel.prietoalhambra@ndorms.ox.ac.uk

