## [Peer Review File · Nature Communications]

Title: Thrombosis and thrombocytopenia after vaccination against and infection with SARS-CoV-2 in the United KingdomREVIEWER COMMENTS

Reviewer #1 (Remarks to the Author): 
I think this paper is worth to be published after major revise.

Reviewer #2 (Remarks to the Author):

Burns et al. present an analysis of primary care data and seeked to measure incidences of VTE, ATE other thromboses and / or thrombocytopenia in temporal relationship with SARS-COV-2 vaccinations. Overall they found low incidences of thrombotic events and thrombocytopenia after SARS COV-2 vaccination, which is somewhat reassuring because it supports the positive risk/benefit ratio of vaccination. The study is of interest, however, as the authors claim in the limitations, has potential weaknesses and sources of bias.

General remark:

1. Based on the indirect comparison, the risk increase reported for the different vaccines appear very minor. Therefore, any comparisons between the vaccines need careful interpretation because variances in the populations and the historical comparators can lead to those observed differences.
2. Pavord et al recently documented 220 cases with vaccine induced thrombotic thrombocytopenia after Chadox1 vaccination in the UK. These patients were particularly younger in agreement to studies from other countries. The EMA also provides a benefit/risk estimate for TTS/VITT regarding vector based vaccines and state that especially younger persons (below 50) are at risk. The authors concede, that the observational period involved mainly older patients who received Chadox1. This could have lead to a substantial risk of missing a safety signal for TTS in younger recipients of Chadox-1.
3. An update of the analysis involving also a time period, were a higher proportion of young recipients were vaccinated seems appropriate.
4. An analysis combining Thrombocytopenia and thrombosis could be added in the figures comparing the vaccines.

Reviewer #3 (Remarks to the Author): 
See attachment.

REVIEWER COMMENTS

Reviewer #1 (Remarks to the Author):

I think this paper is worth to be published after major revise.

NCOMMS-21-33049-TI appreciate the opportunity to review an interesting article submitted to Nature Communications. Burn et al. reported important observations using a large-scale cohort of COVID-19 vaccination. Basically, this reviewer thinks this article provides valuable information to the researchers and physicians who are fighting against COVID-19 in the front line. Since the discussion is relatively simple and short, I ask for some requests for the authors described below.

1. I understand the superiority of this cohort study over the case-series reports. On the other hand, could you please mention the advantage of this study over the reporting system such as the Vaccine Adverse Event Reporting System (VAERS)-based study?

- ➔ **Spontaneous reporting systems such as VAERS, while extremely useful for identifying adverse events after vaccination, do not account for the background rates of events. That is, the events that would be expected to be seen in a population in the absence of any vaccinations. Our study, which uses routinely collected data, includes a general population cohort to estimate these background rates of events. This allows to assess whether the events observed after vaccination are greater than would be expected.**
- ➔ **We have expanded on this in the discussion section of the paper (lines 352 to 362).**

2. How do we interpret the different mechanisms of thrombosis between the ChAdOx1 and BNT162b2 vaccines? It seems that both vaccines slightly increase the incidence of thrombotic events. However, as the SIRs of thrombosis with thrombocytopenia were 1.09 and 0.86, respectively, the mechanisms should be different between the two vaccines. VITT (TTS) is obviously the result of immune reactions but the incidence is quite low. Do you think both vaccines are thrombogenic and the immune reaction-based thrombosis, i.e., TTS occurs almost only after the vaccination with ChAdOx1? If so, what are the underlying mechanism of thrombosis without thrombocytopenia in each vaccine? Is it a common mechanism between ChAdOx1 and BNT162b2? 3. Was the severity different between thromboses after the ChAdOx1 vaccine and those of the BNT162b2 vaccine? The increased risk of thrombosis after BNT162b2 vaccination has not been paid much attention probably because most of the cases were not very severe. In contrast, the thrombosis after ChAdOx1 vaccination especially TTS is often very severe. I know that is not the focus of this report, but the readers want to know how does thrombosis occurs after the vaccination with mRNA vaccine and what are the differences from that of the vectored vaccine.

- Although we cannot delineate these various considerations with the routinely collected data we are using, we have expanded in the discussion to comment further on the mechanisms between the adverse events described (lines 318 to 326).

Minor

1. I suggest to add "in the United Kingdom" to the title.

- We have updated the title accordingly

2. Line 88, 'also' is inappropriate.

- We have removed this

Reviewer #2 (Remarks to the Author):

Burns et al. present an analysis of primary care data and sought to measure incidences of VTE, ATE other thromboses and / or thrombocytopenia in temporal relationship with SARS-COV-2 vaccinations. Overall they found low incidences of thrombotic events and thrombocytopenia after SARS COV-2 vaccination, which is somewhat reassuring because it supports the positive risk/benefit ratio of vaccination. The study is of interest, however, as the authors claim in the limitations, has potential weaknesses and sources of bias.

General remark:

1. Based on the indirect comparison, the risk increase reported for the different vaccines appear very minor. Therefore, any comparisons between the vaccines need careful interpretation because variances in the populations and the historical comparators can lead to those observed differences.

- We agree that any relative risk increases we do see are very small in absolute terms. Including the COVID-19 cohort does, we hope, help to contextualise these results. We have also placed further emphasis on standardised event differences where discussing our results.

2. Pavord et al recently documented 220 cases with vaccine induced thrombotic thrombocytopenia after Chadox1 vaccination in the UK. These patients were particularly younger in agreement to studies from other countries. The EMA also provides a benefit/risk estimate for TTS/VITT regarding vector based vaccines and state that especially younger

persons (below 50) are at risk. The authors concede, that the observational period involved mainly older patients who received Chadox1. This could have lead to a substantial risk of missing a safety signal for TTS in younger recipients of Chadox-1.

→ **In the updated manuscript we have an updated study population with many more younger people now included. We have also now added Figure 4 with results stratified by age group.**

3. An update of the analysis involving also a time period, were a higher proportion of young recipients were vaccinated seems appropriate.

→ **We have updated on our analysis with additional data, with a greater proportion of younger recipients of a vaccine.**

4. An analysis combining Thrombocytopenia and thrombosis could be added in the figures comparing the vaccines.

→ **We have added results for thrombosis and thrombocytopenia to Figure 2 which presents the overall results. Given their rarity, we are not though able to include them in the results stratified by age group (Figure 3 and 4).**

Reviewer #3 (Remarks to the Author):

Thrombosis and thrombocytopenia after vaccination against and infection with SARS-CoV-2: a population-based cohort analysis This large study from the UK by Edward Burn et al. estimates incidence rates of venous thrombosis, arterial thrombosis, thrombocytopenia, and thrombosis with thrombocytopenia in cohorts of 1.9 million people vaccinated with ChAdOx1 (Oxford-AstraZeneca), 1.7 million people vaccinated with BNT162b2 (Pfizer–BioNTech), and 0.3 million people infected with SARS-CoV-2 and compares them with the incidence rates before the pandemic among 2.3 people from the general population. Because much concern has arisen regarding thrombotic events reported among individuals vaccinated with adenovirus-based vaccines, these results are extremely important to put the observations into the correct perspective and to understand whether there was an EXCESS risk among vaccinated persons. The study follows an excellent pre-specified protocol agreed in advance with the study funder, the European Medicines Agency. Similar protocol is also used in respective studies in other European regions, which in the future improves the possibility of comparing the findings of several independent studies and evaluating the role of chance findings.

→ **We thank the reviewer for this positive summary of our paper.**

The incidence of venous thromboembolism (VTE) during the 28 days following vaccination with any of the vaccines was similar to that in the background population, while there was an 8-fold excess in the cohort of persons with SARS-Cov-2 infection. For pulmonary embolism (PE), the SIRs for the vaccinated groups were 21-23% elevated, and in the cohort of persons with SARS-Cov-2 infection it was 15-fold. The risk of thrombocytopenia was increased by 25% after vaccination with ChAdOx1, but not at all increased after vaccination with BNT162b2, while the risk of immune thrombocytopenia was about two times more common than in the background population after vaccination with both ChAdOx1 and BNT162b2. Because the outcomes are very rare, the relative risk estimates should also be explained in terms of absolute excess risk. The authors give the standardized event differences in a Table but do not comment on them at all in the text. I would give such estimates for the main excess findings in the main text and in the Abstract and express them as number of excess cases per 100,000 vaccinated persons instead of the difficult standardized event differences (e.g., 0.0001%).

→ **We agree that emphasising absolute risks as well as relative risks is particularly important in this study. We have now placed further emphasis on standardised event differences throughout.**

The discussion correctly states the potential biases are small and, if there are any, they tend to raise the risk estimates in the vaccinated cohorts. It would be good to highlight in the conclusion that these are rather too high than too low.

→ **Although we agree some biases may lead to overestimates, others may result in underestimates. For example, for other vaccines there is often a concern around “healthy vaccinee bias”, whereby vaccinees are people more likely to adhere to the health recommendations (e.g. less likely to smoke, etc). Although given the high uptake of COVID-19 vaccines this may well be less of a concern, we cannot from the data at hand disentangle possible sources of bias to say in which direction they would be overall.**

Figures: Figure 2 tries to show too much information and is hard to read. The short y axes make it difficult to see whether there are differences in incidence rates between the categories. The values shown on the y axis (e.g., 64-256-1024) are not reader-friendly, either, and the scales are not comparable between different outcomes (same height may mean a 64- or a16-fold variation in rates). This Figure could be presented as a web attachment in a size that makes it easy to see the values, preferably on a log scale with standard values such as 10-20-50-100-200-500 etc.

→ **We have edited this figure, removing the stratification by sex.**
→ **We do agree that the different scales mean that care needs to be taken when interpreting the figure, but given the large differences in incidence rates between events a shared scale, we think, makes it harder to interpret.**

Figure 3 includes so much information that it is hard read. It is also unnecessary because the same results are nicely shown in Table 2

- **Although this figure does show information from Table 2, we think it can still give a visual summary of our study results.**

REVIEWERS' COMMENTS

Reviewer #1 (Remarks to the Author):

I do not have any further comments.

Reviewer #2 (Remarks to the Author):

My concerns were adequately addressed, thank you.